# Customer tolerance in homestays: The influence of interpersonal interaction and motivation Attribution

Huiling Zhou[1], Longfang Huang[2]*, Yu Guo[3]*, Yajun Jiang[3], Ke Wu[1]*

1 School of Economics & Management, Hunan University of Science and Engineering, Hunan, Yongzhou, China, 2 College of International Tourism and Public Administration, Hainan University, Haikou, Hainan, China, 3 College of Tourism & Landscape Architecture, Guilin University of Technology, Guilin, Guangxi, China

* gluthlf@qq.com (LFH); glutgy@163.com (YG); huse_chn@163.com (KW)

## Abstract

Customer tolerance behavior actively sustains harmonious host–guest relationships and strengthens homestay reputations. Drawing on social cognition theory and attribution theory, this study investigates how interpersonal interaction shapes customer tolerance behavior in homestay services and examines whether stay duration moderates these effects. Using survey data from 322 homestay customers collected via the Credamo platform, we test the mechanisms linking interaction quality, motivational attribution, and tolerance. Our findings reveal that interpersonal interaction significantly enhances customer tolerance behavior. Specifically, interaction encourages customers to attribute altruistic rather than self-interested motives to hosts, and these attributions mediate the effect of interaction on tolerance. Although stay duration amplifies the direct effect of interaction on tolerance, it does not significantly moderate the link between interaction and motivational attribution. These findings clarify the psychological processes driving tolerance in homestay contexts and highlight the importance of cultivating positive host–guest interactions to build service resilience.

## Introduction

In today's service economy, tolerance has become a critical behavioral response that reflects how customers manage service shortcomings [1,2]. Tolerance behavior demonstrates customers' patience and forgiveness toward service failures and functions as a key psychological indicator of service evaluation [2,3]. Although some customers respond to service deficiencies with complaints or negative word-of-mouth [1,4,5], tolerance allows them to restore psychological balance and creates opportunities to repair relationships [6]. Several notable scholars have examined tolerance in financial services, healthcare, retail, and hospitality [7–11]. Yet research on tolerance in homestays remains limited, even though this setting emphasizes personalized hospitality and interpersonal engagement more than standardized service systems [12].

**Data availability statement:** All relevant data are within the manuscript and its Supporting Information files.

**Funding:** This research was supported by the National Natural Science Foundation of China [Grant No. 72462013].

**Competing interests:** The authors have declared that no competing interests exist.

Homestays typically operate on a smaller scale [13] and involve frequent, direct host-guest interactions. Unlike traditional hotels, they rely on informal exchanges, emotional connections, and opportunities to engage with local culture and traditions [14–16]. These unique characteristics suggest that tolerance in homestays depends not only on perceived service quality but also on interpersonal dynamics and cultural expectations. In East Asian contexts, especially, where relational orientation and emotional reciprocity hold strong value [17,18], tolerance functions both as a behavioral adjustment and as a cultural expression. However, tourism and hospitality studies have yet to fully explain how host-guest interactions translate into tolerance behavior [19–22].

To fill this gap, we employed attribution theory and social cognitive theory. Attribution theory explains how customers' perceptions of service providers' motives shape their emotional and behavioral responses [23–25]. When guests interpret hosts' behavior as altruistic, they feel gratitude and act more cooperatively [23]. In contrast, egoistic attributions reduce tolerance and weaken relationships [25]. Social cognitive theory highlights the interplay of personal, behavioral, and environmental factors [26]. In homestays, stay duration represents an important environmental factor: shorter stays often generate surface-level impressions, while longer stays allow deeper observation and more complex attributions [27]. Emerging evidence suggests that a three-day threshold may mark the transition from superficial judgments to more elaborate cognitive evaluations [28,29].

Building on these insights, we systematically examine the psychological processes underlying customer tolerance in homestays. Specifically, we addressed the following research questions:

RQ1: Does interpersonal interaction between homestay hosts and guests significantly enhance customers' tolerance behavior?

RQ2: Does motivational attribution mediate the relationship between interpersonal interaction and tolerance behavior?

RQ3: Does stay duration moderate the relationships between interpersonal interaction, motivational attribution, and tolerance behavior?

This study makes three key contributions. Theoretically, it integrates social cognitive theory and attribution theory to explain how interpersonal interactions foster tolerance, extending their application within tourism contexts. It also identifies stay duration as a boundary condition, highlighting the temporal dimension of cognition and behavior in homestays. Practically, it provides actionable insights for homestay operators, emphasizing the importance of nurturing emotional connections and managing guest expectations across varying lengths of stay. By clarifying these mechanisms, we advance both academic understanding and managerial practice in homestay services.

### Theoretical basis and research hypotheses

**Social cognitive theory.** Albert Bandura's social cognitive theory asserts that people learn not only from direct experience but also from observation, imitation, and social interaction within their environment [26]. The theory emphasizes that during

social interactions, individuals perceive, interpret, and internalize others' behaviors and attitudes, which in turn shape their own behavioral patterns. Besides, environmental influences, personal cognition, and observable behavior form a dynamic, reciprocal system in which individuals process stimuli, update their mental representations, and adapt their responses accordingly [26].

Furthermore, several researchers have widely applied social cognitive theory in organizational and consumer behavior, including studies on employees' moral disengagement [30], value co-creation among club members [31], and sustainable purchase intentions [32]. This study, therefore, applies the theory to the homestay context, treating interpersonal interaction as a key external factor influencing customers' cognitive and behavioral responses. It explores how host–guest interactions shape tolerance behavior and contextualizes social cognitive theory in micro-level tourism settings. By introducing stay duration as a moderating variable, the study highlights how the length of stay affects cognitive depth and attributional judgments, which in turn shape the relationship between interpersonal interaction and tolerance behavior. This approach underscores the temporal dimension as a critical factor in consumer behavior and extends the applicability of social cognitive theory within tourism interactions.

**Attribution theory.** Heider first introduced Attribution Theory to explain how individuals interpret the causes of behavior [33]. Kelley later expanded the theory, arguing that people rely on information about consistency, distinctiveness, and covariation to evaluate social behaviors and determine their underlying causes. These evaluations lead to relatively stable attributions regarding external events or others' actions [34]. The theory underscores that individuals' interpretations of others' behaviors strongly shape their own attitudes and behavioral responses [35]. Since attribution is a subjective cognitive process, observers' motivations, attentional focus, and information-processing modes can generate different attributions for [36]identical behaviors [36].

Intriguingly, some notable researchers have widely applied the Attribution Theory in tourism studies, particularly in visit intention [37], loyalty [38], purchase intention [39], and moral emotions or avoidance behaviors [40]. While these studies clarify attribution's role in shaping tourists' emotions, attitudes, and behaviors, the theory fundamentally assumes that individuals evaluate others' motives to form interpretations of behavior. These interpretations, in turn, influence their trust, attitudes, and behavioral decisions [41]. Hence, this study extends Attribution Theory by applying it to homestay services, examining how customers' attributions of hosts' motives trigger distinct emotional and behavioral responses within micro-level tourism interactions.

**Interpersonal interaction and tolerance behavior.** Interpersonal interaction refers to the process by which individuals achieve mutual recognition and understanding through emotional communication and engagement [42]. In the context of this study, it captures the direct, face-to-face dynamics through which homestay hosts and guests build emotional bonds and foster understanding. Prior research confirms that such host–guest interactions significantly shape tourist behavior: they encourage civic engagement [7], strengthen destination affinity, and increase the likelihood of recommendations [43]. Similarly, tourist-to-tourist interactions can influence future behavioral intentions [44] and promote active participation [45]. Social cognition theory highlights the reciprocal influence among individuals, their environment, and behavior, suggesting that people adapt their cognition and actions in response to environmental cues [46,47]. Accordingly, customer behavior in homestay contexts may depend on the quality of their interactions with hosts.

Tourists often choose homestays not only for rest but also for distinctive services, memorable experiences [48,49], and meaningful connections with like-minded individuals [50]. When guests perceive interpersonal exchanges as rooted in cultural traditions and aligned with their values, they tend to respond with greater tolerance [50]. Interpersonal engagement adds unique value to homestays by compensating for functional limitations [51,52], while encouraging cultural exchange and emotional connection [53]. This dynamic allows relationships to evolve from initial unfamiliarity to friendship, fostering spiritual and social fulfillment [54]. A supportive interaction atmosphere not only provides memorable travel experiences but also enhances trust, commitment, and mutual support [55,56]. Based on this reasoning, we hypothesize that:

H1: Interpersonal interaction has a significant positive impact on customer tolerance behavior.

**Interpersonal interaction and motivation attribution.** Interpersonal interaction and motivation attribution represent central elements of cognitive processing. Observers may focus on different aspects of a situation, which leads them to form distinct attributions for the same behavior [36]. In service settings, motivation attribution refers to consumers' judgments about whether service behaviors stem from altruism (genuine concern for others) or egoism (organizational self-interest) [57]. Attribution theory maintains that individuals actively interpret social environments to explain causality [58]. In homestays, customers often evaluate hosts' intentions based on how they perceive and experience interactions. When hosts consistently demonstrate sincerity, enthusiasm, and attentiveness, customers tend to attribute their actions to altruistic motives. Conversely, when interactions appear calculated or transactional, customers often interpret them as egoistically motivated. Such attributions shape customers' emotional experiences, enhance delight, and increase perceived care [16,59,60].

Contemporary research findings suggest that people display altruistic motives in part to gain social approval [61], and these tendencies vary depending on the nature of social interaction [62,63]. Thus, customer-to-customer engagement also strengthens altruistic behavior [64]. Nevertheless, some scholars argue that egoism underlies all human behavior [65,66]. From this perspective, homestay hosts may strategically use interpersonal interaction as a marketing tool that leverages emotional dynamics for organizational gain [67]. In homestay services, warm and attentive hosts can build emotional connections that reduce egoistic attributions and improve customer perceptions [16]. Group-level emotional exchanges may further dampen selfish tendencies [68]. Consequently, customers' emotional involvement in host–guest interactions likely shape their motivational attributions. We therefore propose that:

H2a: Interpersonal interaction has a significant positive effect on customers' altruistic motivation attribution.

H2b: Interpersonal interaction has a significant negative effect on customers' egoistic motivation attribution.

**Motivation attribution and tolerance behavior.** Motivation attribution strongly influences customer tolerance, defined as the willingness to remain patient and forgiving when service quality falls below expectations [3,69]. In this study, tolerance behavior refers to guests' acceptance of inconveniences during their homestay service consumption. Attribution Theory argues that different motivational attributions drive distinct behavioral outcomes [70,71]. For example, Carlson's work shows that perceiving others' behaviors as altruistic increases prosocial behavior [72]. In tourism contexts, tourists' attributions of destinations' motives significantly affect their attitudes and behavioral intentions: altruistic motives foster trust, repeat visitation, and willingness to travel [73]. By contrast, egoistic motives often encourage self-interested behavior, with individuals adjusting their responses to maximize personal benefit [74]. Customers who perceive companies as self-serving usually respond negatively [25].

In homestays, guests who attribute host behaviors to altruism often demonstrate greater empathy and tolerance. They better understand service challenges and respond with patience. However, guests who interpret interactions as egoistic may view hosts as manipulative or untrustworthy [66]. This reduces prosocial behaviors and erodes trust and tolerance. Hence, motivational attribution shapes customers' behavioral responses to interpersonal dynamics. We, therefore, hypothesize that:

H3a: Altruistic motivation attribution has a significant positive effect on customer tolerance behavior.

H3b: Egoistic motivation attribution has a significant negative effect on customer tolerance behavior.

**The mediating role of motivation attribution.** Motivational attribution performs a crucial mediating function across relational dynamics. Studies show that motivation attribution links collectivism with corporate social responsibility (CSR) cognition, establishing a positive correlation [75]. Additionally, altruistic attribution also mediates the relationship between consumers' CSR perceptions and purchase intentions: when consumers attribute company actions to altruistic motives, they report stronger purchase intentions [76]. Similarly, altruistic attribution explains how sustainable product lines offered by fast-fashion retailers enhance legitimacy, as consumers perceive such actions as genuinely altruistic [77]. Other factors, including time and intensity, indirectly shape egoistic attributions, which in turn affect consumers' attitudes and purchasing decisions [78].

This evidence underscores the value of motivation attribution as a mediating variable and highlights its influence on individual behavior. Drawing on these pathways, we argue that motivation attribution mediates the relationship between host–guest interpersonal interaction and customer tolerance behavior in homestay services. Specifically, customers demonstrate tolerance when they interpret host behavior as altruistically motivated, but they show reduced tolerance when they view host actions as egoistically motivated. Thus, motivation attribution functions as a key psychological mechanism linking interpersonal interaction with tolerance. Based on this reasoning, we propose the following hypotheses:

H4a: Customers' altruistic attribution to interpersonal interaction mediates the relationship between interpersonal interaction and tolerance behavior.

H4b: Customers' egoistic attribution to interpersonal interaction mediates the relationship between interpersonal interaction and tolerance behavior.

**Moderation of stay duration.** Stay duration refers to the time between a guest's official check-in and check-out. Homestay stays can range from one or two nights to more than three nights, but research identifies "three days" as a threshold that marks the transition from short-term visitor to familiar guest [28,29].

In homestay services, interpersonal interaction serves as both a core experiential element and a foundation for customers' motivational attributions and behavioral responses. Bandura's social cognitive theory posits that individuals acquire social cues through observation and interaction, integrating these experiences into attributions and behavioral judgments [26]. As service experiences accumulate, hosts better understand customer preferences and provide more personalized services [79]. Longer stays facilitate repeated, meaningful exchanges that allow guests to refine or reassess their interpretations of host motives [80]. These interactions provide richer evidence for distinguishing between genuine altruism and strategic commercial intent. By contrast, shorter stays limit exposure and encourage customers to interpret interactions as transactional, reinforcing egoistic attributions. Based on this reasoning, we propose the following hypotheses:

H5a: Stay duration significantly moderates the relationship between interpersonal interaction and altruistic motivation attribution.

H5b: Stay duration significantly moderates the relationship between interpersonal interaction and egoistic motivation attribution.

Stay duration also shapes customers' tolerance of service deficiencies and their behavioral responses. Social cognitive theory emphasizes that ongoing interactions foster role identity and emotional attachment, which influence behavioral intentions [26]. The findings of prior empirical research further show that the effect of stay duration varies across destination [81], with longer stays generally strengthening destination benefits [82]. Guests who stay more than three days typically build familiarity and emotional bonds, making them more forgiving of minor service shortcomings. In contrast, short-term guests often treat the experience as transactional, responding more critically to service gaps. Moreover, the distinctive interpersonal nature of homestays enables guests to develop cultural insight and emotional connection over time. As these interactions deepen, customers' engagement may shift from transactional exchanges toward relational and identity-based experiences, which foster customer citizenship behaviors [83]. Accordingly, we hypothesize that:

H6: Stay duration significantly moderates the relationship between interpersonal interaction and tolerance behavior.

Consequently, Fig 1 presents the conceptual framework illustrating how host–guest interpersonal interaction influences customer tolerance behavior in homestay services.

## Data source

**Questionnaire design and scale selection.** We determined the sample size for this study. First, G*Power software estimated that a minimum of 210 participants would provide sufficient statistical power (effect size = 0.25, α = 0.05, power = 0.95) [84]. Second, the actual effective sample size of 412 participants exceeded this requirement, ensuring robust analysis. The Academic Committee of the School of Tourism and Landscape Architecture at Guilin University

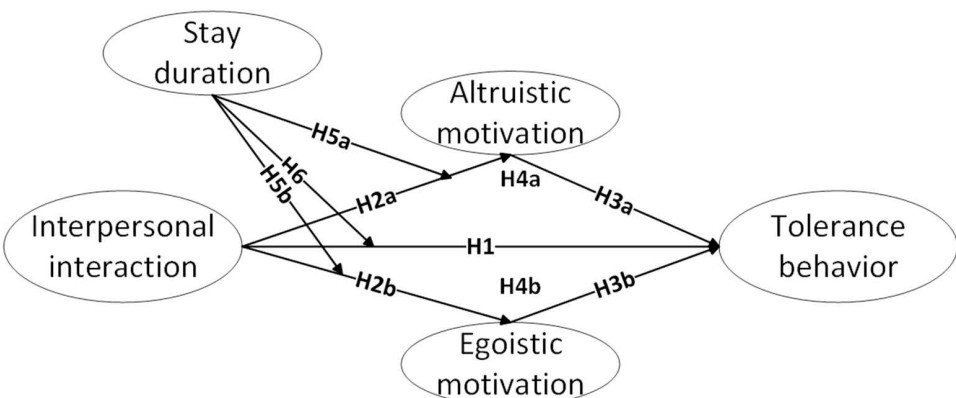

**Fig 1. A conceptual model diagram depicting the effect of interpersonal interaction on tolerance behavior.**

of Technology approved the study. During data collection, researchers presented participants with an overview at the questionnaire's opening, explained relevant details verbally, and obtained informed consent after assuring confidentiality.

The questionnaire consisted of three sections. The first section gathered information on stay duration and distance. The second section measured interpersonal interaction, egoistic and altruistic motivation, and tolerance behavior using established scales. Specifically, we assessed interpersonal interaction with a 4-item scale adapted from Chen et al. [85], measured altruistic and egoistic motivations with a 6-item scale adapted from Rifon et al. [86], and evaluated tolerance behavior with a 3-item scale adapted from Assiouras et al. [87]. We measured stay duration following Jacobsen et al. [88]. To control for demographic influences, we included gender, age, education, occupation, and monthly income [89]. We made minor wording adjustments to align with the homestay context.

Next, we translated all scales from validated English versions into Chinese using a back-translation process. Tourism and hospitality researchers reviewed and refined the draft questionnaire. A pilot test with 90 university students and teachers who had recent homestay experience prompted minor revisions. These steps confirmed the questionnaire's face validity. Also, all constructs used a 7-point Likert scale ranging from 1 (strongly disagree) to 7 (strongly agree). The final section collected demographic characteristics.

**Pre-survey and sample collection.** To minimize social desirability bias, we informed participants that the survey was anonymous, responses would be used only for academic purposes, and accuracy would not be evaluated. We first conducted a pre-survey at a university in Guilin, China, from January 4 to January 7, 2022. Students and teachers with homestay experience returned 90 questionnaires, of which 74 were valid. Exploratory factor analysis showed eigenvalues above 1 for interpersonal interaction, egoistic attribution, altruistic attribution, and tolerance behavior, with factor loadings from 0.649 to 0.916 and no cross-loading. Cronbach's alpha values exceeded 0.8 for all variables, indicating strong internal consistency. The Kaiser–Meyer–Olkin (KMO) values for variables ranged from 0.733 to 0.801, with an overall scale KMO of 0.841, confirming reliability for formal investigation.

Furthermore, the formal survey, administered via the Credamo platform, targeted Chinese users with a homestay experience in the past year. Participants provided informed consent before completing the questionnaire. From February to March 2022, the platform collected 386 responses, of which 322 were valid, producing an effective response rate of 83.41%.

Table 1 presents respondent characteristics. Female participants comprised 56.5% of the sample, while males accounted for 43.5%. Most respondents were under 42 years old, representing the post-80s generation. Educational attainment was largely at bachelor's degree level or higher. Occupations spanned multiple sectors. Regarding monthly

**Table 1. Sample Characteristics (n = 322).**

| Variable | Attribute | Frequency | % | Variable | Attribute | Frequency | % |
|---|---|---|---|---|---|---|---|
| Gender | Male | 140 | 43.5 | Occupation | Student | 46 | 14.3 |
| | Female | 182 | 56.5 | | Enterprise and public institution | 85 | 26.4 |
| Age | 18-22 years | 28 | 8.7 | | Professionals | 34 | 10.6 |
| | 23-32 years | 169 | 52.5 | | Agriculture, forestry, animal husbandry and fishery | 11 | 3.4 |
| | 33-42 years | 101 | 31.4 | | Commercial and service industry personnel | 120 | 37.3 |
| | 43-52 years | 21 | 6.5 | | Production and transportation-related personnel | 13 | 4.0 |
| | 53 years and above | 3 | 0.9 | | Self-employed | 5 | 1.6 |
| Education | | | | | Retiree | 3 | 0.9 |
| | Junior high school | 6 | 1.9 | | Others | 5 | 1.6 |
| | Senior high school | 14 | 4.3 | Monthly income | Below 2,000 yuan | 27 | 8.4 |
| | Junior college | 40 | 12.4 | | 2000-2999 yuan | 17 | 5.3 |
| | Bachelor | 238 | 73.9 | | 3000-4999 yuan | 34 | 10.6 |
| | Master | 22 | 6.8 | | 5000-7999 yuan | 89 | 27.6 |
| | Doctor | 2 | 0.6 | | More than 8,000 yuan | 155 | 48.1 |

income, 48.1% earned more than 8,000 yuan, 27.6% earned 5,000–7,999 yuan, 10.6% earned 3,000–4,999 yuan, 5.3% earned 2,000–2,999 yuan, and 8.4% earned less than 2,000 yuan. These demographics align with the platform's user profile and mirror the 2020 China Homestay Industry Research Report [90], confirming the sample's representativeness.

## Results and analysis

### Common method bias test

We conducted a common method bias test that employed methodology to evaluate the presence of common method variance (CMV). The primary approach utilized the Harman one-factor test, a widely established technique for assessing CMV [91]. A result below 50% suggests an absence of significant CMV issues [92]. In this study, the first factor value was 28.314%, falling below both the 50% threshold and the more stringent 40% benchmark, thus providing robust evidence against the presence of common method variance in the dataset.

### Reliability and validity analysis

The reliability and validity analysis of this study entailed the use of a Cronbach's alpha test to assess the internal reliability of each scale prior to testing the measurement model. As presented in Table 3, all coefficient values exceeding 0.70 were deemed acceptable. Equally, the confirmatory factor analysis (CFA) evaluated the validity of the measurement model. As shown in Table 3, the path coefficients exceeded the cut-off point of 0.5. Our measurement model, therefore, demonstrated good fit with the dataset based on criteria established by Hu and Bentler [93] (see Table 2). The combined reliability (CR) and the average variance extracted (AVE) of each variable surpassed the 0.7 and 0.5 critical cut-off points, respectively, indicating high scale reliability and validity.

**Correlation analysis.** Table 4 presents the correlation coefficients among variables. The results revealed significant correlations between the variables. Besides, the study employed the variance inflation factor (VIF) to examine multicollinearity effects (Table 4). The VIF values between the acceptable parameters, i.e., from 1.289 to 2.250, indicate no multicollinearity issues.

**Table 2. Fitting index of the model.**

| Models | χ²/df | RMSEA | GFI | NFI | IFI | TLI | CFI |
|---|---|---|---|---|---|---|---|
| standard | 1-3 | <0.08 | >0.9 | >0.9 | >0.9 | >0.9 | >0.9 |
| CMV | 1.988 | 0.055 | 0.958 | 0.956 | 0.978 | 0.962 | 0.977 |
| Measurement | 2.725 | 0.073 | 0.929 | 0.923 | 0.950 | 0.933 | 0.949 |

**Table 3. Reliability and validity analysis.**

| Variables/ items | path coefficient | CR | AVE | Cronbach 's α |
|---|---|---|---|---|
| *interpersonal interaction* | | 0.764 | 0.448 | 0.759 |
| The communication between the homestay staff and me is comprehensive and warm | 0.689 | | | |
| The homestay staff receive me like friends, and the atmosphere is relaxing | 0.613 | | | |
| Through the introduction of the homestay host, I have a deeper understanding of the local folk culture and experience real life | 0.684 | | | |
| Through the introduction of the homestay host, I meet other friends | 0.688 | | | |
| *altruism* | | 0.783 | 0.550 | 0.746 |
| The homestay staff keep a good relationship with me because it really cares about the guests | 0.861 | | | |
| The homestay staff keep a good relationship with me to care for the interests of the guests | 0.709 | | | |
| The homestay staff keep a good relationship with me to provide a welcoming atmosphere | 0.636 | | | |
| *egoism* | | 0.913 | 0.777 | 0.912 |
| The homestay staff keep a good relationship with me in order to get more customers | 0.852 | | | |
| The homestay staff keep a good relationship with me because they ultimately care about their own interests | 0.885 | | | |
| The homestay staff keep a good relationship with me in order to get good reviews | 0.906 | | | |
| *tolerance behavior* | | 0.818 | 0.602 | 0.804 |
| If the service of this homestay is not satisfactory, I would stand it | 0.745 | | | |
| If this homestay staff make a mistake in the service process, I am willing to wait patiently | 0.692 | | | |
| If I need to wait longer than I expected, then I am willing to adapt | 0.879 | | | |

Note: CR = Composite Reliability; AVE = Average Variance. Extracted.

## Main Effect Test

To ascertain the main effects of the model, the demographic factors influencing variables such as age, gender, occupation, and education were controlled and specified as covariates. Furthermore, we conducted multiple regression analysis using SPSS 23.0 (see Table 5) with altruistic attribution, egoistic attribution, and tolerance behavior set as dependent variables. The results indicate that interpersonal interaction exhibits a significant positive influence on altruistic attribution, while showing a substantial negative effect on egoistic attribution, thereby supporting Hypothesis 2 (a-b). Moreover, interpersonal interaction was found to significantly enhance tolerance behavior, confirming Hypothesis 1. Also, we find that egoistic attribution negatively impacts tolerance behavior, while altruistic attribution positively influences tolerance behavior, thereby supporting Hypothesis 3 (a-b).

**Table 4. Correlation analysis.**

| variable | interpersonal interaction | egoism | altruism | tolerance behavior | VIF |
|---|---|---|---|---|---|
| interpersonal interaction | 1 | | | | 1.920 |
| egoism | −0.127[a] | 1 | | | 1.137 |
| altruism | 0.688[b] | −0.292[b] | 1 | | 2.250 |
| tolerance behavior | 0.315[b] | −0.272[b] | 0.450[b] | 1 | 1.289 |

Note: a indicates $p < 0.05$, b indicates $p < 0.01$.

**Table 5. Analysis of main effects.**

| Variables | Egoistic attribution | Altruistic attribution | Tolerance behavior | | |
|---|---|---|---|---|---|
| | beta coefficients | beta coefficients | beta coefficients | beta coefficients | beta coefficients |
| Gender | 0.28 | −0.036 | −0.173 | −0.123 | −0.122 |
| Age | 0.355** | −0.027 | 0.028 | 0.091 | 0.083 |
| Education | 0.05 | 0.022 | −0.093* | −0.084* | −0.098 |
| Occupation | 0.021 | 0.073 | −0.076 | −0.073 | −0.112* |
| Monthly income | −0.297*** | 0.055 | 0.144* | 0.091 | 0.081 |
| Interpersonal interaction | −0.246* | 0.852*** | 0.5*** | 0.456*** | 0.03 |
| Egoistic attribution | | | | −0.178*** | −0.115** |
| Altruistic attribution | | | | | 0.519*** |
| $R^2$ | 0.079 | 0.484 | 0.128 | 0.174 | 0.246 |
| $\Delta R^2$ | 0.061 | 0.474 | 0.111 | 0.155 | 0.227 |
| F | 4.488*** | 49.19*** | 7.703*** | 9.428*** | 12.78*** |
| $\Delta F$ | 4.051* | 241.143*** | 27.712*** | 17.377*** | 30.121*** |

Note: * $p < 0.05$, ** $p < 0.01$ and *** $p < 0.001$.

## Mediating effect analysis

Using the SPSS PROCESS plug-in command (Model 4), a bootstrapping method with 5,000 resamples examined the mediating effects of egoistic and altruistic attributions. The results revealed in Table 6 indicate an indirect effect of altruistic attribution exists between interpersonal interaction and tolerance behavior (0.2935), with a confidence interval excluding zero. Similarly, the indirect effect of egoistic attribution (0.0257) revealed a confidence interval not encompassing zero. These findings confirm motivation attribution as a significant mediator between interpersonal interaction and tolerance behavior, thereby supporting Hypothesis H4 (a-b).

## Moderation effect analysis

To examine the moderating role of stay duration, this study analyzes egoistic attribution, altruistic attribution, and tolerance behavior. We used these dependent variables to centrally process demographic variables, interpersonal interaction, stay duration, and interaction terms, before employing a hierarchical regression model for interaction term analysis. The results exhibited in Table 7 shows that the interaction of interpersonal interaction and stay duration on egoistic attribution (β=−0.05, p>0.05) and altruistic attribution (β=0.047, p>0.05) were not significant. While the interaction between interpersonal interaction and stay duration demonstrated a significant positive effect on tolerance behavior (β=0.735, p<0.001), indicating the stay duration's significant positive moderating effect on the relationship that exists between interpersonal

**Table 6. Analysis of mediating effects.**

| path | Effect coefficient | standard error | 95% confidence interval | |
|---|---|---|---|---|
| | | | lower limit | Upper limit |
| Interpersonal interaction —— tolerance behavior | 0.2930 | 0.095 | 0.3132 | 0.6870 |
| Interpersonal interaction—— altruistic attribution | 0.6651 | 0.0549 | 0.7442 | 0.9601 |
| Interpersonal interaction —— egoistic attribution | −0.1152 | 0.1224 | −0.4872 | −0.0055 |
| altruistic attribution——tolerance behavior | 0.4413 | 0.0919 | 0.4071 | 0.7686 |
| egoistic attribution——tolerance behavior | −0.2228 | 0.0426 | −0.2616 | −0.0938 |
| Interpersonal interaction —— altruism——tolerance behavior | 0.2935 | 0.0526 | 0.1971 | 0.4032 |
| Interpersonal interaction —— egoism ——tolerance behavior | 0.0257 | 0.0153 | 0.0012 | 0.0604 |

**Table 7. Test of moderating effects.**

| Variables | Egoism attribution | Altruism attribution | Tolerance behavior |
|---|---|---|---|
| | beta coefficients | beta coefficients | beta coefficients |
| Gender | 0.277 | −0.035 | −0.185 |
| Age | 0.391** | −0.042 | 0.005 |
| Education | 0.058 | 0.019 | −0.091* |
| Occupation | 0.038 | 0.067 | −0.062 |
| Monthly income | −0.251** | 0.036 | 0.103 |
| Interpersonal interaction | −0.215 | 0.842*** | 0.548*** |
| Stay duration | −0.826** | 0.349** | 0.712*** |
| Interpersonal interaction×Stay duration | −0.05 | 0.047 | 0.735*** |
| $R^2$ | 0.114 | 0.501 | 0.194 |
| $\triangle R^2$ | 0.091 | 0.488 | 0.173 |
| F | 5.033*** | 39.291*** | 9.416*** |
| $\triangle F$ | 0.038 | 0.167 | 14.399*** |

Note: * $p < 0.05$, ** $p < 0.01$ and *** $p < 0.001$.

interaction and tolerance behavior. Consequently, Hypotheses 5a and 5b were not supported. However, Hypothesis 6 was supported.

To verify the robustness of the moderating effect, stay duration was categorized into two groups: "short-term tour" for stays of 3 days or less [29], coded as 0 (sample size = 141), and "medium to long-term tour" for stays exceeding 3 days, coded as 1 (sample size = 181). This binary classification of stay duration enabled further analysis using the SPSS PROCESS plug-in command in Model 1 with Bootstrapping using 5,000 resamples to verify the moderating effect of stay duration on the relationship between interpersonal interaction and tolerance behavior. The results revealed that the interaction term between interpersonal interaction and stay duration had a significant positive impact on tolerance behavior (β=0.037, p < 0.001). This is consistent with the regression results, which support Hypothesis 6, thus confirming the robustness and reliability of our findings. To illustrate this moderating effect more clearly, the relationship between interpersonal interaction perception and tolerance behavior was analyzed separately for customers with stays of three days or less and for those exceeding three days. The results presented in Fig 2 of the moderating effect diagram revealed no significant effect for short-term stays (β= −0.048, p > 0.05). However, longer stays had a significant effect (β= 0.687, p < 0.001).

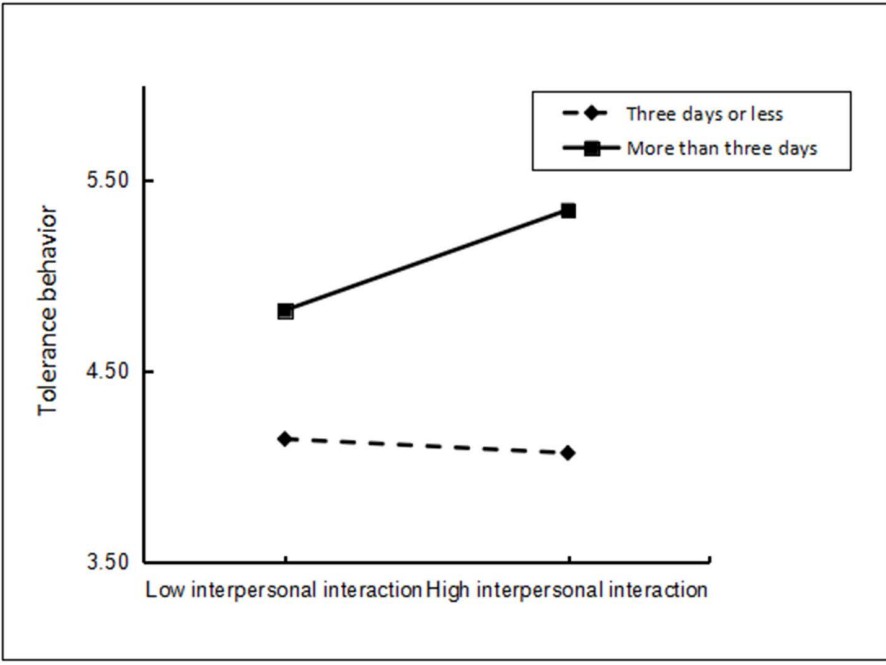

**Fig 2. Moderating effect of stay duration.**

## Discussion

A growing number of scholars have increasingly examined customer tolerance behavior in tourism consumption, yet its underlying mechanisms remain insufficiently explored. This study clarifies these mechanisms by testing a moderated mediation model that links interpersonal interaction in homestays to customer tolerance behavior. Motivation attribution functions as the core mediator, and stay duration operates as a key boundary condition. The empirical results strongly support the proposed hypotheses. Interpersonal interaction exerts a significant positive influence on customer tolerance behavior through motivation attribution, and longer stays strengthen this direct effect.

First, the findings confirm that interpersonal interaction significantly shapes customer tolerance behavior (H1). This outcome echoes prior studies highlighting interpersonal interaction's role in driving tourist behavioral outcomes. For instance, Chen et al. found that host-guest interaction affects tourists' behavioral intentions and attitudes [85], while Lin et al. found positive effects on satisfaction and revisit intentions [20]. Thus, the present results extend these insights by demonstrating that richer interpersonal interaction in homestays fosters cultural exchange and emotional connection [60]. Such exchanges satisfy tourists' expectations of social connection and experiential value [48,49]. When service flaws arise, customers respond with greater patience and understanding. These findings advance theoretical understanding by positioning interpersonal interaction as a central driver of tolerance in homestay contexts.

Second, the analysis reveals that interpersonal interaction shapes customers' attribution of motives, enhancing altruistic attributions while diminishing egoistic ones (H2a, H2b). This pattern supports Kelley's attribution theory [34], which emphasizes that individuals interpret others' motives in social contexts to guide their emotional and behavioral responses. In homestays, warm hospitality, genuine service, and authentic emotional expression encourage customers to perceive hosts as altruistically motivated—concerned with customer well-being and quality service [16]. Such prosocial attributions foster trust and strengthen emotional ties [23]. By contrast, interactions perceived as formulaic or profit-driven—such as scripted greetings or aggressive upselling—prompt egoistic attributions [25]. Therefore, customers then discount the

sincerity of the exchange, reducing its perceived value. Motivational attribution thus operates as a critical cognitive mechanism through which interpersonal interactions influence behavioral outcomes.

Third, customers' motivational attributions directly shape their tolerance behavior (H3a, H3b). Specifically, attributing altruistic motives increases tolerance, while egoistic attributions reduce it. Customers who perceive genuine care form stronger emotional bonds and are more forgiving of service lapses, consistent with emotional intensification theory. Tolerance, in this case, reflects a prosocial behavioral response to perceived sincerity [16]. Conversely, egoistic attributions generate suspicion and defensiveness, diminishing satisfaction and tolerance. Prior studies support this pattern. Carlson and Zaki argue that egoistic attributions limit emotional engagement and behavioral flexibility [72], while Choy et al. note that utilitarian interpretations undermine trust and promote critical responses [66]. These findings highlight the psychological pathway by which motive perception translates into behavioral tolerance.

Fourth, the results confirm that motivation attribution mediates the effect of interpersonal interaction on tolerance behavior (H4a, H4b). This finding reinforces prior evidence that consumers' interpretations of others' motives shape their attitudes and behaviors [23,24]. Interpersonal interaction alone does not dictate tolerance; rather, customers' cognitive appraisals of motives determine whether interaction fosters patience and understanding. When customers attribute altruistic intent, they develop identification and emotional bonding [59,60], which enhances tolerance of service shortcomings. By contrast, egoistic attributions block trust formation, even when interactions appear superficially positive. This mediating role of attribution underscores the importance of motivational transparency and sincerity in homestay services. The study thereby strengthens attribution theory's relevance in tourism and service research.

Fifth, the results highlight the moderating effect of stay duration (H5a, H5b, H6). Longer stays significantly strengthen the direct link between interpersonal interaction and tolerance behavior, though not the indirect pathway through motivation attribution. This pattern aligns with Haverila & Haverila's observation that trip duration shapes satisfaction pathways [80]. Specifically, customers who stay three days or more develop deeper impressions of interpersonal interaction. Sustained engagement allows them to build stronger emotional bonds, which in turn increase tolerance. However, the absence of moderation on the attribution pathway suggests that motive evaluation occurs early in the customer experience. Once customers form an initial attribution, they focus less on reevaluating motives and more on assessing overall service quality during extended stays. Thus, longer stays amplify the emotional and relational dimensions of interaction but do not alter the initial cognitive attribution process.

Collectively, these findings clarify the mechanisms that drive tolerance behavior in homestay settings. Interpersonal interaction emerges as a central antecedent, motivation attribution explains its psychological effects, and stay duration strengthens its behavioral impact. This integrated perspective enriches both theoretical and practical understanding of customer tolerance in tourism consumption.

### Theoretical implications

This study establishes interpersonal interaction as a critical foundation of customer tolerance behavior, thereby extending research on customer citizenship behavior in homestay services. Previous scholarship has typically treated interpersonal interaction as part of host–guest exchanges, emphasizing functional communication and service procedures. Such a perspective neglects the deeper emotional processes embedded in interpersonal relationships and their impact on tourists' psychology and behavior. Homestays, unlike standardized hotels, operate on a smaller scale and involve frequent, personalized exchanges that foster emotional care and intimacy [13]. Positive interpersonal interactions enhance customers' emotional experiences, cultivate belonging and engagement, and strengthen their acceptance of local culture, customs, and social values [14–16]. By highlighting the emotional dimension, this study addresses the limitations of functionally oriented research. It positions emotion as a psychological pathway that shapes customer attitudes and behavioral intentions, enriching the theoretical understanding of tolerance behavior.

Furthermore, this research also advances the literature by incorporating motive attribution into the study of tourist behavior. Earlier studies focused on motive attribution in shaping purchase intentions and brand attitudes [76,78], but

overlooked its role in service encounters such as homestays. In immersive, highly interpersonal contexts, customers interpret service providers' motives as either altruistic or egoistic, and these judgments substantially influence their behavior. The study empirically validates motive attribution as a mediator, revealing a sequential mechanism: interpretation of others' behavior→cognitive–emotional processing→behavioral response. This mechanism provides a more nuanced explanatory framework and shifts the discussion of tourism psychology from outcome-based to process-based perspectives, while extending attribution theory's application in relational service contexts.

Over and above, this study highlights the moderating role of stay duration, which earlier research often considered only as a determinant or mediator of customer behavior [27]. By positioning stay duration as a moderator, the study demonstrates how time conditions shape the strength of interpersonal interaction's effect on tolerance behavior. Rather than the conventional 1-day threshold [94], this research employs a 3-day threshold, which is more representative of typical short-term travel. Our findings show that guests who stay longer form stronger emotional bonds with hosts and display greater tolerance of service flaws. This focus on time as a boundary condition addresses a gap in behavioral research and introduces the temporal dimension as a key moderator. The results not only enrich theory but also offer practical insights for designing stage-specific service strategies in homestays.

### Practical implications

The findings generate actionable guidance for homestay managers seeking to enhance customer tolerance. Because interpersonal interaction significantly boosts tolerance, managers should embed a relationship-driven service model at the core of operations, prioritizing emotional engagement and trust-building.

At the service management level, cultivating authentic and emotionally rich host–guest interactions prove essential. Since altruistic attributions strengthen tolerance, managers should encourage hospitality practices perceived as genuine, such as attentive communication and sincere care. In contrast, mechanical or overtly promotional behaviors risk generating egoistic attributions, reducing tolerance. Managers should also recognize that extended stays magnify the effect of interpersonal interaction, making sustained relationship-building with long-term guests especially valuable.

On top of that, staff training should include attribution awareness. Employees need to signal altruistic intent consistently through genuine hospitality and meaningful cultural sharing, moving beyond standardized scripts. Homestays can also cultivate an altruistic image at the brand level by highlighting community involvement, origin stories, or nonprofit partnerships. Such initiatives reinforce customer trust and emotional connection. To further enhance tolerance, operators could design empathy-building experiences, such as behind-the-scenes tours, immersive local activities, or storytelling sessions that reveal operational challenges. These approaches help customers contextualize service imperfections and respond with greater patience.

Additionally, this study also identifies "three days" as a psychological threshold for tolerance development. Managers can leverage this insight by structuring marketing and service strategies around extended stays. Three targeted practices stand out: (1) incentivize bookings through promotions like "stay three nights, get one free"; (2) ensure high-quality interpersonal interaction and memorable experiences during the first two days to encourage longer stays; and (3) address service failures promptly through recovery measures such as upgrades, complimentary experiences, or discounts. Together, these strategies align service design with customer psychology, fostering resilience and loyalty.

### Conclusion

This study developed and tested a moderated mediation model to explain how interpersonal interaction influences customer tolerance behavior in homestays. Motivation attribution emerged as a mediating mechanism, while stay duration functioned as a key boundary condition. The results confirm three contributions. First, interpersonal interaction exerts a significant positive effect on customer tolerance, addressing the first research question. Second, motivation attribution mediates this relationship, validating the second research question. Third, stay duration moderates the direct link between interpersonal interaction and tolerance behavior, though it does not moderate the indirect attribution pathway—providing

partial support for the third research question. Collectively, these findings extend theoretical understanding of tolerance behavior in hospitality and offer actionable insights for practice.

Despite the numerous contributions of this study, several limitations open avenues for future research. First, this study analyzed tolerance primarily from the customer perspective. Future work should integrate host perspectives to capture dyadic dynamics and broaden the analysis of civic behaviors. Second, while the study identified a partial mediating role for attribution, additional mediators should be tested to strengthen the framework. Beyond stay duration, moderators such as personality, cultural values, or trust could refine the understanding of boundary conditions [95,96]. Furthermore, the study's Chinese context limits generalizability; future research should compare urban versus rural settings and cross-cultural differences. Notably, reliance on online surveys may limit ecological validity. Employing a mixed-methods approach that integrates online and on-site data could enhance representativeness and robustness.

We, therefore, recommend that homestay managers prioritize authentic interpersonal interaction and emphasize altruistic service motives to strengthen customer tolerance. Taken together, managers should invest in staff training that fosters genuine hospitality, design cultural experiences that build empathy, and strategically encourage stays beyond three days to deepen host–guest bonds. By aligning service practices with customer psychology, homestays can enhance tolerance, improve satisfaction, and build long-term loyalty.

## Supporting information

**S1. Data.**
(PDF)

## Author contributions

**Data curation:** Longfang Huang, Yu Guo.

**Formal analysis:** Yu Guo.

**Funding acquisition:** Yajun Jiang.

**Investigation:** Yu Guo.

**Resources:** Longfang Huang.

**Software:** Longfang Huang.

**Supervision:** Huiling Zhou.

**Validation:** Yajun Jiang, Ke Wu.

**Visualization:** Huiling Zhou, Ke Wu.

**Writing – original draft:** Huiling Zhou, Longfang Huang, Yu Guo, Yajun Jiang.

**Writing – review & editing:** Ke Wu.

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
