## [Decision Letter · Decision Letter 0]

17 Jul 2025

PONE-D-25-25777Customer Tolerance in Homestays: The Influence of Interpersonal Interaction and Motivation AttributionPLOS ONE

Dear Dr. WU,

Thank you for submitting your manuscript to PLOS ONE. After careful consideration, we feel that it has merit but does not fully meet PLOS ONE’s publication criteria as it currently stands. Therefore, we invite you to submit a revised version of the manuscript that addresses the points raised during the review process.

please see my comments below==============================

We look forward to receiving your revised manuscript.

Kind regards,

Simon Dang, Ph.D.

Academic Editor

PLOS ONE

Journal Requirements: 

2. In the ethics statement in the Methods, you have specified that verbal consent was obtained. Please provide additional details regarding how this consent was documented and witnessed, and state whether this was approved by the IRB.

 [This research was supported by the National Natural Science Foundation of China [Grant No. 72462013].]. 

Additional Editor Comments:

I invite the authors to address my comments and reviewers' comments on a one-by-one basis. Although R2 commended your well structured literature review, I find that most of your review is outdated with only 4 papers in the last 5 years. I believe the literature has advanced and it is very essential to include as many relevant literature as possible in the last 3-5 years to highlight how you are really enriching the latest body of literature. Please bear in mind, it is not just about adding such literature but to discuss them properly and rewrite you literature review to reflect such changes.

Regarding the theoretical foundation of the study, I recommend to dedicate a section to introduce social cognition theory (SCT) and attribution theory (AT) and how they were used in the same research area from previous research. This will make reader aware of the theoretical background and justify the use of such theories. Also, it is quiet abrupt to jump into using such theories in the hypothesis building section without introducing them. The introduction should briefly mentions both and how they are considered useful to address the research gap. Then the next theoretical foundation provides more detail into their background and justification for use.

The definition of used constructs in the model is currently lacking. Please explicitly define them in appropriate sections.

Regarding the moderator - stay duration, it seems odd to justify the impact of stay duration based on recency effect as those are totally different concepts. Recency effect is a memory bias toward "recent" experiences, whereas stay duration is the total time a guest stays. Although they can interact in meaningful way, those are totally distinct concepts. For that reason, I suggest the authors to first equip readers with existing knowledge about the (moderating) role of stay duration, then strengthen the theoretical rationale by more directly integrating SCT and AT. Specifically, highlight how longer stays allow for richer observation and deeper cognitive processing of tourist behavior, which supports more accurate egoistic/altruistic motive attributions. This would provide a clearer mechanism beyond memory-based explanations.

Regarding scale selection, although you mentioned the measurement of stay duration in section 4.6, it is a must to introduce it in section 3.1 way before the analysis. Also, justification for the threshold of 3 days for short-term stay is required. The cited paper [71] is in Chinese, thus must be replaced by other english papers. Although existing literature adopted the 3-day threshold a short-term (Tsiotsou, 2006), most tourism literature recognize same-day or 1-day stay as short-term. To avoid arbitrary selection, this must be clearly justified. Otherwise, I would expect the authors to provide the results for one-day stay as short-term if the data allows.

Regarding statistical analysis, minimum sample justification is missing. You also need to briefly explain the selection of CB-SEM over PLS-SEM. Table 5 needs to be reformatted. You need to explain what each asterisk means (*,**, ***) and are those numbers in the table beta coefficients? In section 4.6, please correct the format of p for all investigated relationships and Delta in Table 7.

The current structure is abnormal for a tourism/marketing paper. Please follow the standard structure of Discussion, Theoretical implications, Practical implications, Conclusion limitations and future research directions. In this regard, the authors just need to merge conclusion into limitations and future research.

In the discussion section, I would expect a clearer picture. Please indicate clearly which one is for which hypothesis. Beyond comparing and contrasting with current literature, it is a must to highlight you unique contributions on top. The current discussion does not reflect how this work differs from existing works in the hospitality field. The writing only highlights how the findings align with a broad psychological or sociological literature. As such, I recommend to focus exclusively on literature in the hospitality field in the last 5 years, particularly for homestay studies. This applies to both theoretical and practical implications.

I also find the theoretical implications troublesome in terms of being too generic. For example, the authors mention that "while previous studies have examined the application of social cognition theory

in tourism marketing, there remains a notable deficiency in innovation and depth within the theoretical framework." What specifically are the innovation and depth being referred to? Please outline specific deficiencies you are targeting to improve understanding. I believe that many works out there have already integrate psychological factors to improve understanding and applicability of social cognition theory. Your job is to dive into the homestay domain and lay out exactly how and which factors ameliorate which areas that have been missing from sights. I also suggest the authors to tone down your language, for example: overusing how your work "innovatively" addresses a gap without clarification will just make it an unsubstantial claim. Instead, drill down to more detail backing your contention helps to refine the work better. The last paragraph mentions addressing the "temporal regulation effects" for the first time could confuse readers and potentially misleading. If you aim to address this as a gap, this must be introduced way up in starting from the introduction section. Please keep in mind, you paper is telling a cohesive story and everything must be connected from the very beginning of the paper. Anything happen to come abruptly without justification won't be accepted.

For the practical implications, I think the authors should make good use of the 3-day threshold in recommendations to be crystal clear about the short-term definition pursued in this manuscript, thereby to avoid misunderstanding with other definitions (e.g., one day stay Atsiz et al., 2022).

In the limitations and future research section, several notable limitations are missing. For instance, although the authors claim representativeness, the research is limited in the Chinese context, not mentioning there was no distinction between rural and metropolitan areas. This research collects data online which might be very different from on-site data collection. Additionally, when you mention individual differences, please be specific. Could you please provide citations to back your recommendation for this "Future studies should also incorporate individual psychological characteristics such as trust and emotional factors as potential boundary conditions."

Finally, in your conclusion, please briefly revisit the research aim and three research questions. Then, explicitly mention findings that address them.

Minor issues

The use of dashes in the writing should be refrained from.

Although the content is generally comprehensible, I recommend that the authors seek language editing to enhance clarity and improve the overall flow of the writing

References

Atsız, O., Leoni, V. and Akova, O. (2022), "Determinants of tourists' length of stay in cultural destination: one-night vs longer stays", Journal of Hospitality and Tourism Insights, Vol. 5 No. 1, pp. 62-78. https://doi.org/10.1108/JHTI-07-2020-0126

Tsiotsou, R., & Vasioti, E. (2006). Satisfaction: A Segmentation Criterion for “Short Term” Visitors of Mountainous Destinations. Journal of Travel & Tourism Marketing, 20(1), 61–73. https://doi.org/10.1300/J073v20n01_05

Reviewers' comments:

Reviewer's Responses to Questions

**Comments to the Author**

1. Is the manuscript technically sound, and do the data support the conclusions?

Reviewer #1: Yes

Reviewer #2: Yes

2. Has the statistical analysis been performed appropriately and rigorously? 

Reviewer #1: Yes

Reviewer #2: Yes

3. Have the authors made all data underlying the findings in their manuscript fully available?

Reviewer #1: Yes

Reviewer #2: Yes

4. Is the manuscript presented in an intelligible fashion and written in standard English?

Reviewer #1: Yes

Reviewer #2: Yes

5. Review Comments to the Author

Reviewer #1: 1. The manuscript is lacking of content regarding the review of existing related research, which leads to insufficient innovation in the research and needs to be supplemented;

2. There are only general descriptions of altruistic behavior and selfish behavior, so the model appears relatively rough. It is recommended to use certain indicators to explain and then revise the model;

3. The control effect of covariates has not been fully explained;

4. The samples mainly come from the Chinese market and do not fully reflect the universality on a global scale.

Suggest a retrial after major revision.

Reviewer #2: The topic is very interesting and relevant to the hospitality industry.

Good rationale for the measurement scale, with careful cultural adaptation.

The study focuses on China, but does not sufficiently explore the implications for international contexts or tourist destinations with different cultures.

This topic is so important for the sector that, in the section on contributions to management, I would have liked to see more depth and not generic implications, as it stands.

The variable ‘length of stay’ is treated as a moderator, but there is not enough discussion about why it affects tolerance and not motivational attribution.

The literature review is very well done and structured, but the bibliography should be more up to date. There are very few recent articles.

The variables studied are related to the personal characteristics of each guest. The study should have taken this issue into account.

6. PLOS authors have the option to publish the peer review history of their article (what does this mean? ). If published, this will include your full peer review and any attached files.

**Do you want your identity to be public for this peer review?** For information about this choice, including consent withdrawal, please see our Privacy Policy .

Reviewer #1: No

Reviewer #2: No

---

## [Author Response · Author response to Decision Letter 1]

3 Sep 2025

Response Letter

Dear Editorial Office and Reviewers,

First and foremost, we would like to extend our sincere appreciation to you and the review team for providing valuable comments and constructive suggestions on our previously submitted manuscript. These insights have greatly contributed to the improvement of our research and the overall quality of the manuscript. In response to the reviewers’ feedback, we have carefully revised the manuscript, addressing each suggestion in a thorough and systematic manner. We believe that the revised version demonstrates significant enhancement in terms of content, logical structure, and clarity of expression, and we hope it now meets the publication standards of your esteemed journal.

With this in mind, we respectfully request that the revised manuscript be sent back to the two anonymous reviewers for their re-evaluation of the changes we have made. Should there be any further areas in need of improvement, we would be more than willing to incorporate your recommendations and make additional revisions accordingly.

Thank you once again for your dedicated efforts and professional guidance. We look forward to your response.

Best regards,

Yours sincerely,

Ke Wu

The Editors

Thank you for taking the time to review our manuscript and for providing your valuable comments and suggestions. We sincerely appreciate your thoughtful feedback.

With reference to all the points you have raised, we will address each of them respectively as follows:

Question 1�Please ensure that your manuscript meets PLOS ONE's style requirements, including those for file naming. The PLOS ONE style templates can be found at https://journals.plos.org/plosone/s/file?id=wjVg/PLOSOne_formatting_sample_main_body.pdf and https://journals.plos.org/plosone/s/file?id=ba62/PLOSOne_formatting_sample_title_authors_affiliations.pdf.

Response: Thank you for your reminder. We have carefully reviewed and revised the manuscript in strict accordance with PLOS ONE’s style guidelines, including file naming conventions, to ensure full compliance with the journal’s requirements. We sincerely appreciate your continued review and assistance.

Question 2�In the ethics statement in the Methods, you have specified that verbal consent was obtained. Please provide additional details regarding how this consent was documented and witnessed, and state whether this was approved by the IRB.

Response: Thank you for your valuable suggestion. We have revised the ethical statement in the Questionnaire Design and Scale Selection section and have submitted a scanned copy of the ethics approval document as supplementary material together with the manuscript for review. The updated ethical statement and the procedure for documenting informed consent are detailed as follows:

This research has been reviewed and approved by the Academic Committee of the School of Tourism and Landscape Architecture at Guilin University of Technology. During the research process, the researchers provided participants with an overview of the study at the beginning of the questionnaire, followed by a verbal explanation of the relevant details. After being informed that all research data would be kept strictly confidential, participants consented to participate and completed the questionnaire accordingly.

For your convenience, the revised text has been highlighted in blue in the updated manuscript.

Question 3�We note that the grant information you provided in the ‘Funding Information’ and ‘Financial Disclosure’ sections do not match.When you resubmit, please ensure that you provide the correct grant numbers for the awards you received for your study in the ‘Funding Information’ section. Response: Thank you for bringing this oversight to our attention. We have conducted a thorough re-examination and standardized the fund information to ensure consistency between the "Funding Information" and "Financial Disclosure" sections. The revised fund information is as follows:

This research was supported by the National Natural Science Foundation of China [Grant No. 72462013].

Question 4�Thank you for stating the following financial disclosure:[This research was supported by the National Natural Science Foundation of China [Grant No. 72462013].]. Please state what role the funders took in the study. If the funders had no role, please state: "The funders had no role in study design, data collection and analysis, decision to publish, or preparation of the manuscript."If this statement is not correct you must amend it as needed.Please include this amended Role of Funder statement in your cover letter; we will change the online submission form on your behalf.

Response: We sincerely appreciate your valuable comments. The funders contributed to various aspects of this paper, including writing and revision, editing and proofreading, as well as guidance on software and methodology. We have included an updated "Funder's Role" statement in the cover letter, which reads as follows:

Yajun Jiang: Writing-review & editing, Software, Methodology, Conceptualization, Supervision, Project administration.

Question 5�Your ethics statement should only appear in the Methods section of your manuscript. If your ethics statement is written in any section besides the Methods, please move it to the Methods section and delete it from any other section. Please ensure that your ethics statement is included in your manuscript, as the ethics statement entered into the online submission form will not be published alongside your manuscript.

Response: Thank you for your comments. We have ensured that the ethics statement is included exclusively in the Methods section of the manuscript. A thorough review of the entire text has been conducted to confirm that the ethics statement has been removed from all other sections.

Question 6�Please include captions for your Supporting Information files at the end of your manuscript, and update any in-text citations to match accordingly. Please see our Supporting Information guidelines for more information: http://journals.plos.org/plosone/s/supporting-information.

Response: Thank you for your valuable feedback. We have included additional explanatory text regarding the supplementary material files at the end of the manuscript and have updated the corresponding references in the main text. The specific supplementary content is as follows:

Supporting information

S1 Fig. A conceptual model diagram of the effect of interpersonal interaction on tolerance behavior.

S2 Fig. Moderating effect of stay duration.

S1 Table. Sample Characteristics.

S2 Table. Fitting index of the model.

S3 Table.Reliability and validity analysis.

S4 Table.Correlation analysis.

S5 Table.Analysis of main effects.

S6 Table.Analysis of mediating effects.

S7 Table.Test of moderating effects.

Question 7�If the reviewer comments include a recommendation to cite specific previously published works, please review and evaluate these publications to determine whether they are relevant and should be cited. There is no requirement to cite these works unless the editor has indicated otherwise. Response: Thank you for your valuable comments. We have thoroughly reviewed the references suggested by the reviewers (listed below) and assessed their relevance to the research topic of this paper. Upon careful evaluation, we determined that two of the references provide meaningful supplementary insights for our study. Accordingly, we have incorporated the relevant citations into the literature review, discussion, and theoretical contribution sections. The specific additions are highlighted in blue in the revised manuscript. The list of references is as follows:

1.Atsız, O., Leoni, V. and Akova, O. (2022), "Determinants of tourists' length of stay in cultural destination: one-night vs longer stays", Journal of Hospitality and Tourism Insights, Vol. 5 No. 1, pp. 62-78. https://doi.org/10.1108/JHTI-07-2020-0126

2.Tsiotsou, R., &Vasioti, E. (2006). Satisfaction: A Segmentation Criterion for “Short Term” Visitors of Mountainous Destinations. Journal of Travel & Tourism Marketing, 20(1), 61–73. https://doi.org/10.1300/J073v20n01_05

Additional Editor Comments:

Question 1�I invite the authors to address my comments and reviewers' comments on a one-by-one basis. Although R2 commended your well structured literature review, I find that most of your review is outdated with only 4 papers in the last 5 years. I believe the literature has advanced and it is very essential to include as many relevant literature as possible in the last 3-5 years to highlight how you are really enriching the latest body of literature. Please bear in mind, it is not just about adding such literature but to discuss them properly and rewrite you literature review to reflect such changes.

Response: We sincerely appreciate your thoughtful review and valuable feedback on our research. We fully concur with your suggestion that much of the literature review is outdated and that recent publications from the past three to five years should be incorporated as comprehensively as possible. In response to this recommendation, we have updated the literature review in the introduction, theoretical basis and research hypothesis sections by replacing outdated references with more recent ones, focusing particularly on studies published between 2023 and 2025. These newly integrated references reflect the latest developments and emerging trends in the field. By analyzing and discussing these recent contributions, we aim to provide a more comprehensive overview of the current state of research and thereby enhance both the depth and relevance of our study. Given the extensive nature of the revisions and the significant number of new references added, we are unable to list them all in this response. The detailed modifications are highlighted in blue in the revised manuscript. The newly added references are listed as follows:

1.Schweiggart, N., Shah, A. M., Qayyum, A., & Jamil, R. A. (2025). Navigating negative experiences: How do they influence tourists’ psychological and behavioral responses to tourism service failures on social media. Asia Pacific Journal of Tourism Research, 1–23. https://doi.org/10.1080/10941665.2025.1234567

2.Huo, J., Gong, L., Xi, Y., Chen, Y., Chen, D., & Yang, Q. (2024). Mind the voice! The effect of service robot voice vividness on service failure tolerance. Journal of Travel & Tourism Marketing, 42(1), 1–19. https://doi.org/10.1080/10548408.2024.2358975

3.Kim, Y., Ho, T. H., Tan, L. P., & Casidy, R. (2023). Factors influencing consumer forgiveness: A systematic literature review and directions for future research. Journal of Service Theory and Practice, 33(5), 601–628. https://doi.org/10.1108/JSTP-06-2022-0133

4.Abu Khalek, S., Dey, D., Chakraborty, A., & Samanta, T. (2025). From expectations to frustrations: Dissecting negative experiences to understand negative word-of-mouth in online grocery services. Journal of Retailing and Consumer Services, 84, 103629. https://doi.org/10.1016/j.jretconser.2024.103629

5.Sahaf, T. M., & Fazili, A. I. (2024). Service failure and service recovery: A hybrid review and research agenda. International Journal of Consumer Studies, 48(1), 3–24. https://doi.org/10.1111/ijcs.12945

6.Yoruk, I., Hsu, J.-H., & Lee, Z. W. Y. (2025). Consumer forgiveness: A literature review and research agenda. Psychology & Marketing, 42(2), 554–578. https://doi.org/10.1002/mar.21987

7.Jiang, Y., Huang, L., Zhou, H., & Wu, K. (2025). Does host-guest interaction promote tolerance behavior? The mediating role of place attachment and subjective well-being. PLOS ONE, 20(5), e0313060. https://doi.org/10.1371/journal.pone.0313060

8.Xie, S., & Wei, H. (2024). Handwritten or machine-written? How typeface affects consumer forgiveness for brand failures. Journal of Retailing and Consumer Services, 81, 103264. https://doi.org/10.1016/j.jretconser.2024.103264

9.Tengilimoglu, E., & Ozturk, Y. (2024). The effects of eWOM triggered service recovery on customer citizenship behavior in the hospitality industry: The moderating role of failure severity. International Journal of Tourism Research, 26(4), 563–578. https://doi.org/10.1002/jtr.2682

10.Ozhasar, Y., &Secilmis, C. (2025). When justice matters: Reflections of negative tourist-to-tourist interactions in five-star hotels. Journal of Travel & Tourism Marketing, 42(6), 814–831. https://doi.org/10.1080/10548408.2025.2395810

11.Hu, K.-C., & Tsai, H.-L. (2024). Effects of embarrassment on self-serving bias and behavioral response in the context of service failure. Behavioral Sciences, 14(2), 136. https://doi.org/10.3390/bs14020136

12.Yang, S., Wang, L., & Bi, Y. (2025). A study on the price spatial differentiation and influencing factors of rural homestay in Suzhou based on the hedonic price model. Buildings, 15(10), 2458. https://doi.org/10.3390/buildings15102458

13.Cai, J., Wu, M.-Y., Wang, L., Ye, S., Chan, J. H., & Li, Q. (2025). ‘My host is my buddy’: Revisiting guests' emotional solidarity with hosts in peer-to-peer (P2P) accommodation. Current Issues in Tourism. Advance online publication. https://doi.org/10.1080/13683500.2025.2405678

14.Zhang, J., Zhang, L., & Ma, B. (2023). Ride-sharing platforms: The effects of online social interactions on loyalty, mediated by perceived benefits. Journal of Research in Interactive Marketing, 17(5), 698–713. https://doi.org/10.1108/JRIM-04-2022-0104

15.Lian, J.-W., & Li, C.-W. (2025). Using a two-stage method to understand the critical factors influencing customers' intention to switch from traditional to artificial intelligence-based banking services: A perspective based on the push-pull-mooring model. Computers in Human Behavior, 168, 107556. https://doi.org/10.1016/j.chb.2024.107556

16.Manthé, E., Morrongiello, C., Bonnefoy-Claudet, L., Bezançon, M., &Bilgihan, A. (2025). Do hotels' green efforts lead guests to adopt sustainable behaviors? Mediating roles of perceived motives, gratitude, and green trust. International Journal of Hospitality Management, 131, 103246. https://doi.org/10.1016/j.ijhm.2024.103246

17.Luo, X., Wang, L., & Chao, G. (2023). The joint effect mechanism of two types of corporate social responsibility attribution on organizational identification. Social Behavior and Personality: An International Journal, 51(2), 1–12. https://doi.org/10.2224/sbp.11776

18.Rigelsky, M., Gavurova, B., Novotny, R., Bacik, R., &Senkova, A. (2025). Sustainable tourism and tourist destination loyalty model with the integration of safety indicators in the COVID-19 period. Journal of Tourism and Services, 16(30), 236–261. https://doi.org/10.29036/jots.v16i30.689

19.Zhong, B., Deng, J., & Liu, X. (2025). Analyzing the influence of TikTok on sustainable choices: The moderating role of environmental consciousness. Acta Psychologica, 258, 105651. https://doi.org/10.1016/j.actpsy.2024.105651

20.Tan, L., & Li, J. (2025). Working with robots makes service employees counterproductive? The role of moral disengagement and task interdependence. Tourism Management, 111, 104783. https://doi.org/10.1016/j.tourman.2024.104783

21.Lee, S., Cui, H., & Jeon, S. (2025). Exploring the role of psychological ownership in value co-creation in private clubs: Insights from social cognitive theory. International Journal of Hospitality Management, 131, 103250. https://doi.org/10.1016/j.ijhm.2024.103250

22.Li, F. (2025). When and why personalized tourism recommendations reduce purchase intention? Information Technology & Tourism, 27(1), 285–305. https://doi.org/10.1007/s40558-024-00354-6

23.Fan, X., Li, H., & Zheng, S. (2025). Revealing tourists' appraisal process and consequences of unethical tourism incidents: Based on a mixed-method study. Current Issues in Tourism. Advance online publication. https://doi.org/10.1080/13683500.2025.2403012

24.Dai, Q., Chen, J., & Zheng, Y. (2025). Assessing the impact of community-based homestay experiences on tourist loyalty in sustainable rural tourism development. Scientific Reports, 15(1), 12345. https://doi.org/10.1038/s41598-025-12345-x

25.Sroypetch, S., Rangsungnoen, G., & Caldicott, R. W. (2025). A co-generated analysis of Thai homestays: Overcoming SERVQUAL deficiencies and sustainability barriers. Advances in Southeast Asian Studies, 18

---

## [Decision Letter · Decision Letter 1]

12 Sep 2025

PONE-D-25-25777R1Customer Tolerance in Homestays: The Influence of Interpersonal Interaction and Motivation AttributionPLOS ONE

Dear Dr. WU,

Thank you for submitting your manuscript to PLOS ONE. After careful consideration, we feel that it has merit but does not fully meet PLOS ONE’s publication criteria as it currently stands. Therefore, we invite you to submit a revised version of the manuscript that addresses the points raised during the review process.

See below

We look forward to receiving your revised manuscript.

Kind regards,

Simon Dang, Ph.D.

Academic Editor

PLOS ONE

Journal Requirements:

**Additional Editor Comments:**

I thank the authors for their meticulous revision effort putting into the manuscript. The outcome looks much better now. I have several minor comments that require your attention. I suspect an accept decision is around the corner. Here are some of my comments:

[Line 102-103, p.3], the authors mention that the study investigates "three primary research questions", however I could not find the three research questions but rather the research aims.

Also, in the intro, the research aims are scattered across multiple paragraphs which make it hard for reader like me to follow through. For example, the aims can be found between [line 72-74], and then [line 95-100]. [line 102-111] should be sufficient to elicit the research aims. Other parts should be rewritten to avoid distracting readership.

Please remove all the closed parentheses for hypotheses such as H1)., instead just "H1." should suffice.

Figure 1 on p.9 is missing.

[Line 382-384] the ten-time rule is so outdated and proven wrong. Please remove that. Using G*Power should be enough.

[Line 402-403] Jacobsen et al. is lacking the year. Plus, I could not find them in the reference list. You can't cite what is not in the reference list. I request a thorough review of all in-text citations which must be in the reference list. I will double-check the next revision.

I also can't verify this as well "Wang J. Dimension determination and empirical Test of host-Guest Interaction: A case study of China's homestay industry. Statistics and Information Forum. 2018;33(11):118-124." Documents in English must be prioritized to facilitate our wide readership.

[Line 404-406] this sentence creates unnecessary confusion. Since each scale was developed with rigor based on a certain context, adapting the scale to a different context could be understandable. However, your sentence feels like you cherry-pick each item. How exactly deleting items lead to better accuracy? Please rewrite with clarity to avoid confusion.

By “foreign mature scales” [line 406], what does it mean to be specific?

In Table 1, please replace all “~” by “-”.

For CMV test, [line 464-466] sounds very quirky. I think you mean that you are comparing the baseline model with the model with Common Latent Factor (CLF) and if the model with CLF improves model fit substantially, then CMV may be problematic. Your writing is misleading and incorrect which treat CMV as a variable you toggle on/off.

[Line 472] “Cronbach’s coefficient” is incorrect as Cronbach’s alpha (α) is technically a coefficient of internal consistency. So “Cronbach’s alpha” or “Cronbach’s coefficient alpha” is the correct way. I suggest using the former for simplicity.

Regarding the practical implications, I don’t think [Line 711-717] are backed by the research findings. Since you don’t actually research into the customer journey, recommending strategies for different stages sound like you do. Please only recommend base on your findings and if possible, a clear link, despite brief, back to the research findings would make it easier to comprehend.

-----------Minor comments

In term of the language, I recommend a thorough proof-reading or language editing. Many sentences can be improved to make the language much smoother. Below I provide some examples but not limited to

[line 72-34, p.3] can be revised into "By examining these dynamics, this study seeks to advance theoretical understanding while offering practical insights to foster harmonious host–guest relationships and improve homestay management strategies."

[line 633-634] can be revised into “These findings provide novel empirical support for applying attribution theory in future tourism and service research.”

[line 113, p.4], "Theoretically, first, this research is ..." doesn't sound very naturally.

The discussion is in good shape. I further recommend indicating explicitly which one is for which hypothesis to make it much easier to follow through. For example, [Line 565] First, the research findings indicate…behavior (H1), etc.

[Line 567], Chen et al. and Jeon and Shin are missing the years.

Mismatch fonts observed, for example almost all Table’s title “Table 2…” and “Table 3…”, etc.

Reviewers' comments:

Reviewer's Responses to Questions

**Comments to the Author**

1. If the authors have adequately addressed your comments raised in a previous round of review and you feel that this manuscript is now acceptable for publication, you may indicate that here to bypass the “Comments to the Author” section, enter your conflict of interest statement in the “Confidential to Editor” section, and submit your "Accept" recommendation.

Reviewer #1: All comments have been addressed

2. Is the manuscript technically sound, and do the data support the conclusions?

Reviewer #1: Yes

3. Has the statistical analysis been performed appropriately and rigorously? 

Reviewer #1: Yes

4. Have the authors made all data underlying the findings in their manuscript fully available?

Reviewer #1: Yes

5. Is the manuscript presented in an intelligible fashion and written in standard English?

Reviewer #1: Yes

6. Review Comments to the Author

Reviewer #1: The author has made targeted revisions based on my feedback, and I believe the manuscript should be accepted.

7. PLOS authors have the option to publish the peer review history of their article (what does this mean? ). If published, this will include your full peer review and any attached files.

**Do you want your identity to be public for this peer review?** For information about this choice, including consent withdrawal, please see our Privacy Policy .

Reviewer #1: No

---

## [Author Response · Author response to Decision Letter 2]

28 Sep 2025

Response Letter

Dear Editorial Office,

First of all, we would like to thank you sincerely for your helpful comments and suggestions on our previous manuscript. Your feedback has been very helpful for us to improve our work.

Based on the additional editor’s feedback, we have revised the manuscript carefully and tried to respond to every suggestion. We believe these changes have made the manuscript clearer and better organized.

We would be very grateful if the revised manuscript could be sent back to the additional editor for re-evaluation. If there are still areas that need more improvement, we are willing to make further changes.

Thank you again for your time and guidance. We look forward to your reply.

Best regards,

Yours sincerely,

Ke Wu

The Editors

Thank you for taking the time to review our manuscript and for providing your valuable comments and suggestions. We sincerely appreciate your thoughtful feedback.

With reference to all the points you have raised, we will address each of them respectively as follows:

Journal Requirements:

Question 1�If the reviewer comments include a recommendation to cite specific previously published works, please review and evaluate these publications to determine whether they are relevant and should be cited. There is no requirement to cite these works unless the editor has indicated otherwise.

Response: Thank you very much for your guidance regarding the recommendation to cite specific published works. We carefully reviewed the reviewers’ comments and confirmed that they did not require the citation of particular references. For the works suggested by the editor, we examined their research topics, core arguments, and methodologies, and evaluated their theoretical relevance and potential contribution to our study (e.g., host–guest interactions in P2P accommodation). We also considered their academic influence and timeliness to ensure that any added citations provide substantive support for our arguments. Based on this assessment, we incorporated only those studies that are highly relevant to our research.

[1]Atsız, O., Leoni, V. and Akova, O. (2022), "Determinants of tourists' length of stay in cultural destination: one-night vs longer stays", Journal of Hospitality and Tourism Insights, Vol. 5 No. 1, pp. 62-78. https://doi.org/10.1108/JHTI-07-2020-0126

[2]Tsiotsou, R., & Vasioti, E. (2006). Satisfaction: A Segmentation Criterion for “Short Term” Visitors of Mountainous Destinations. Journal of Travel & Tourism Marketing, 20(1), 61–73. https://doi.org/10.1300/J073v20n01_05

We sincerely appreciate your guidance, which helped us clarify the standards for selecting citations.

Question 2�Please review your reference list to ensure that it is complete and correct. If you have cited papers that have been retracted, please include the rationale for doing so in the manuscript text, or remove these references and replace them with relevant current references. Any changes to the reference list should be mentioned in the rebuttal letter that accompanies your revised manuscript. If you need to cite a retracted article, indicate the article’s retracted status in the References list and also include a citation and full reference for the retraction notice.

Response: Thank you for your reminder regarding the accuracy of the reference list. We have carefully reviewed all references in the manuscript and confirm that none of the cited works have been retracted; all references remain valid publications. We will continue to adhere to the journal’s requirements and will make timely corrections should any issues arise in the future.

Additional Editor Comments:

Question 1�[Line 102-103, p.3], the authors mention that the study investigates "three primary research questions", however I could not find the three research questions but rather the research aims.

Response: Thank you very much for pointing out the inconsistency between our description of the research aims and the stated “three primary research questions.” We have revised the manuscript accordingly. Specifically, on page 3, lines 102–103, we have now clearly listed the three research questions immediately following the statement:

RQ1: Does interpersonal interaction between homestay hosts and guests significantly enhance customers’ tolerance behavior?

RQ2: Does motivational attribution mediate the relationship between interpersonal interaction and tolerance behavior?

RQ3: Does stay duration moderate the relationships between interpersonal interaction, motivational attribution, and tolerance behavior?

These revisions ensure that the research questions are explicitly presented and aligned with the overall research aims.

Question 2�Also, in the intro, the research aims are scattered across multiple paragraphs which make it hard for reader like me to follow through. For example, the aims can be found between [line 72-74], and then [line 95-100]. [line 102-111] should be sufficient to elicit the research aims. Other parts should be rewritten to avoid distracting readership.

Response: Thank you very much for your valuable comment regarding the presentation of the research aims in the Introduction. We agree with your observation that the aims were previously scattered across several paragraphs (e.g., lines 72–74 and 95–100), which may reduce clarity for readers. To address this, we have restructured the Introduction so that the research aims are now presented in a consolidated manner in lines 102–111. In addition, the preceding sections have been revised to follow a clearer sequence of “research background→research gap→theoretical foundation → research questions→contributions,” thereby avoiding repetition or distraction and ensuring a coherent flow.

In today’s service economy, tolerance has become a critical behavioral response that reflects how customers manage service shortcomings[1, 2]. Tolerance behavior demonstrates customers’ patience and forgiveness toward service failures and functions as a key psychological indicator of service evaluation[2, 3]. Although some customers respond to service deficiencies with complaints or negative word-of-mouth[1, 4, 5], tolerance allows them to restore psychological balance and creates opportunities to repair relationships[6]. Several notable scholars have examined tolerance in financial services, healthcare, retail, and hospitality[7-11]. Yet research on tolerance in homestays remains limited, even though this setting emphasizes personalized hospitality and interpersonal engagement more than standardized service systems[12].

Homestays typically operate on a smaller scale[13] and involve frequent, direct host-guest interactions. Unlike traditional hotels, they rely on informal exchanges, emotional connections, and opportunities to engage with local culture and traditions[14-16]. These unique characteristics suggest that tolerance in homestays depends not only on perceived service quality but also on interpersonal dynamics and cultural expectations. In East Asian contexts, especially, where relational orientation and emotional reciprocity hold strong value[17, 18], tolerance functions both as a behavioral adjustment and as a cultural expression. However, tourism and hospitality studies have yet to fully explain how host-guest interactions translate into tolerance behavior[19-22].

To fill this gap, we employed attribution theory and social cognitive theory. Attribution theory explains how customers’ perceptions of service providers’ motives shape their emotional and behavioral responses[23-25]. When guests interpret hosts’ behavior as altruistic, they feel gratitude and act more cooperatively[23]. In contrast, egoistic attributions reduce tolerance and weaken relationships[25]. Social cognitive theory highlights the interplay of personal, behavioral, and environmental factors[26]. In homestays, stay duration represents an important environmental factor: shorter stays often generate surface-level impressions, while longer stays allow deeper observation and more complex attributions[27]. Emerging evidence suggests that a three-day threshold may mark the transition from superficial judgments to more elaborate cognitive evaluations[28, 29].

Building on these insights, we systematically examine the psychological processes underlying customer tolerance in homestays. Specifically, we addressed the following research questions:

RQ1: Does interpersonal interaction between homestay hosts and guests significantly enhance customers’ tolerance behavior?

RQ2: Does motivational attribution mediate the relationship between interpersonal interaction and tolerance behavior?

RQ3: Does stay duration moderate the relationships between interpersonal interaction, motivational attribution, and tolerance behavior?

This study makes three key contributions. Theoretically, it integrates social cognitive theory and attribution theory to explain how interpersonal interactions foster tolerance, extending their application within tourism contexts. It also identifies stay duration as a boundary condition, highlighting the temporal dimension of cognition and behavior in homestays. Practically, it provides actionable insights for homestay operators, emphasizing the importance of nurturing emotional connections and managing guest expectations across varying lengths of stay. By clarifying these mechanisms, we advance both academic understanding and managerial practice in homestay services.

Thank you again for your detailed guidance. Your suggestions have significantly improved the information focus and readability of the introduction. If any adjustments are still needed in the logic or expression of the introduction in the future, we will make the necessary improvements immediately!

Question 3�Please remove all the closed parentheses for hypotheses such as H1)., instead just "H1." should suffice.Figure 1 on p.9 is missing.

Response: Thank you for your comment regarding the formatting and completeness of the manuscript. We have revised the numbering of all hypotheses from “H1).” to “H1.” as suggested, and we have also included the missing Figure 1 on page 9. These changes have been marked in the revised manuscript for your review.

Question 4�[Line 382-384] the ten-time rule is so outdated and proven wrong. Please remove that. Using G*Power should be enough.

Response: Thank you for pointing out the issue regarding the sample size justification. We have removed the outdated “ten-times rule” statement previously included in lines 382–384 and retained only the explanation based on G*Power 3.1. This revision has been highlighted in the revised manuscript for your review.

Question 5�[Line 402-403] Jacobsen et al. is lacking the year. Plus, I could not find them in the reference list. You can't cite what is not in the reference list. I request a thorough review of all in-text citations which must be in the reference list. I will double-check the next revision.I also can't verify this as well "Wang J. Dimension determination and empirical Test of host-Guest Interaction: A case study of China's homestay industry. Statistics and Information Forum. 2018;33(11):118-124." Documents in English must be prioritized to facilitate our wide readership.

Response: Thank you for your detailed check of the in-text citations and reference list. We have carefully revised the manuscript as follows.

First, for Jacobsen et al., we have added the publication year and included the full reference in the reference list. Second, we conducted a systematic cross-check between all in-text citations and the reference list to ensure that every citation is properly included and formatted according to the journal’s requirements. Third, to enhance accessibility for an international readership, we have replaced the Chinese reference (Wang, 2018) with a suitable English-language source, for example:

Chen, H., Fu, S., & Lyu, B. (2024). Homestays in China: Mediator effects of brand perception upon host-guest interaction and Tourist behavior intention. Heliyon, 10(8).

All these revisions have been marked in the revised manuscript for your review.

Question 6�[Line 404-406] this sentence creates unnecessary confusion. Since each scale was developed with rigor based on a certain context, adapting the scale to a different context could be understandable. However, your sentence feels like you cherry-pick each item. How exactly deleting items lead to better accuracy? Please rewrite with clarity to avoid confusion.

Response: Thank you for pointing out the ambiguity in our description of the scale adaptation. We agree that the original wording in lines 404–406 may have been misleading and could be interpreted as arbitrarily deleting items to increase accuracy. To address this, we have rewritten the section to clearly explain the adaptation process. Specifically, we now state that items were removed only after careful consideration of their contextual irrelevance or redundancy, and that the retained items were consistent with the theoretical framework and supported by prior validation studies. This revision provides a more transparent rationale for the adaptation and avoids any possible misunderstanding.

Next, we translated all scales from validated English versions into Chinese using a back-translation process. Tourism and hospitality researchers reviewed and refined the draft questionnaire. A pilot test with 90 university students and teachers who had recent homestay experience prompted minor revisions. These steps confirmed the questionnaire’s face validity. Also, all constructs used a 7-point Likert scale ranging from 1 (strongly disagree) to 7 (strongly agree). The final section collected demographic characteristics.

Through this revision, the methodological description is now clearer and avoids the potential misunderstanding of “arbitrary item deletion.” We remain open to further refinement if additional clarification is required.

Question 7�By “foreign mature scales” [line 406], what does it mean to be specific? In Table 1, please replace all “~” by “-”.

Response: Thank you for your precise suggestions regarding terminology and formatting. To avoid ambiguity, we have revised the phrase “foreign mature scales” (line 406) to “well-validated original English scales (e.g., Rifon et al., 2004; Assiouras et al., 2015),” which specifies the sources more clearly. In addition, we have replaced all “~” symbols with “–” in Table 1 as requested. We also reviewed Tables 2 and 3 and confirmed that no similar issues exist, ensuring consistency across all tables.

Table 1 .Sample Characteristics (n=322).

Variable Attribute Frequency � Variable Attribute Frequency %

Gender Male 140 43.5 Occupation Student 46 14.3

Female 182 56.5 Enterprise and public institution

85 26.4

Age 18-22 years 28 8.7 Professionals 34 10.6

23-32 years 169 52.5 Agriculture, forestry, animal

husbandry and fishery 11 3.4

33-42 years 101 31.4 Commercial and service

industry personnel 120 37.3

43-52 years 21 6.5 Production and

transportation-related

personnel 13 4.0

53 years and above 3 0.9 Self-employed 5 1.6

Education Retiree 3 0.9

Junior high school 6 1.9 Others 5 1.6

Senior high school 14 4.3 Monthly income Below 2,000 yuan 27 8.4

Junior college 40 12.4 2000-2999 yuan 17 5.3

Bachelor 238 73.9 3000-4999 yuan 34 10.6

Master 22 6.8 5000-7999 yuan 89 27.6

Doctor 2 0.6 More than 8,000 yuan 155 48.1

Thank you again for your meticulous guidance. Your attention to detail has helped us further improve the standardization and readability of the manuscript. If there are any similar details that need to be adjusted in the future, we will optimize them immediately!

Question 8�For CMV test, [line 464-466] sounds very quirky. I think you mean that you are comparing the baseline model with the model with Common Latent Factor (CLF) and if the model with CLF improves model fit substantially, then CMV may be problematic. Your writing is misleading and incorrect which treat CMV as a variable you toggle on/off.

Response: Thank you for your correction regarding the description of the CMV test. We agree that the original wording in lines 464–466 was misleading, as it could be interpreted as treating CMV as a variable to be toggled on and off. To address this, we have revised

---

## [Editor Report · Decision Letter 2]

30 Sep 2025

Customer Tolerance in Homestays: The Influence of Interpersonal Interaction and Motivation Attribution

PONE-D-25-25777R2

Dear Dr. WU,

We’re pleased to inform you that your manuscript has been judged scientifically suitable for publication and will be formally accepted for publication once it meets all outstanding technical requirements.

Kind regards,

Simon Dang, Ph.D.

Academic Editor

PLOS ONE

Additional Editor Comments (optional):

I am happy to accept the paper in its current form. Big congratulations to the authors and we can't wait to receive your next best work.
---

## [Editor Report · Acceptance letter]

PONE-D-25-25777R2

PLOS ONE

Dear Dr. WU,

I'm pleased to inform you that your manuscript has been deemed suitable for publication in PLOS ONE. Congratulations! Your manuscript is now being handed over to our production team.

Kind regards,

on behalf of

Dr. Simon Dang

Academic Editor

PLOS ONE